## RESEARCH ARTICLE

# Microtubule lattice defects facilitate spastin-mediated severing

Cordula Reuther[1,*], Paula Santos-Otte[1,*], Rahul Grover[1], Till Korten[1] and Stefan Diez[1,2,3,‡]

## ABSTRACT

The length regulation of microtubules and their organization into complex arrays inside cells occurs through the activity of polymerases and depolymerases, as well as severing enzymes, such as spastin and katanin. Spastin and katanin hexamerize on the microtubule lattice, pull out single tubulin dimers in an ATP-dependent manner and eventually generate internal breaks in the microtubule. Although spastin has been shown to be regulated by post-translational tubulin modifications, the impact of microtubule lattice defects on the severing characteristics of spastin has not been explored. To address this, we prepared GMPCPP-stabilized microtubules with varying defect densities – introduced either through specific polymerization conditions or by end-to-end annealing – for subsequent *in vitro* severing assays. We found that: (1) the presence of defects accelerated the onset of the severing process; and (2) severing was twice as frequent in microtubule segments with defect sites when compared to random lattice segments. However, there was no evidence of preferential binding of spastin to defect sites. We therefore propose a severing mechanism in which defects do not actively promote microtubule severing but, instead, passively contribute to microtubule lattice instability. The defects thus facilitate the severing process by reducing the number of tubulin subunits that must be removed for severing to occur.

KEY WORDS: Microtubule severing, Spastin, Lattice defects

## INTRODUCTION

Microtubule regulators, including severing enzymes such as spastin, katanin and fidgetin organize the microtubule cytoskeleton inside cells by controlling their length and growth dynamics. Severing enzymes use the energy of ATP hydrolysis to extract tubulin subunits from microtubules and, hence, generate internal breaks in the microtubule lattice that lead to severing events (Roll-Mecak and McNally, 2010; Akhmanova and Steinmetz, 2015). Microtubule severing by spastin and katanin plays pivotal roles in different cellular processes such as neurodevelopment (Ahmad et al., 1999; Sherwood et al., 2004), and mitosis and meiosis (Mains et al., 1990; Zhang et al., 2007). Furthermore, spastin contributes to endosomal fission and trafficking and shaping of the endoplasmatic reticulum (Liu et al., 2021; Allison et al., 2013, 2017), as well as controlling

[1]B CUBE – Center for Molecular Bioengineering, TUD Dresden University of Technology, Dresden D-01307, Germany. [2]Cluster of Excellence Physics of Life, TUD Dresden University of Technology, Dresden D-01307, Germany. [3]Max Planck Institute of Molecular Cell Biology and Genetics, Dresden D-01307, Germany. *These authors contributed equally to this work

‡Author for correspondence (stefan.diez@tu-dresden.de)

C.R., 0000-0002-3787-4022; P.S.-O., 0009-0009-4556-0574; R.G., 0000-0002-5373-6578; T.K., 0000-0002-2315-9247; S.D., 0000-0002-0750-8515

lipid droplet dispersion (Tadepalle et al., 2020; Tadepalle and Rugarli, 2021), whereas katanin is heavily involved in ciliogenesis (Dymek et al., 2004) and cell migration (Zhang et al., 2011), and in coordinating the remodeling, orientation and amplification of microtubule arrays in plants (Lindeboom et al., 2013; Zhang et al., 2013). The diverse roles of microtubule severing, which involve both assembly and disassembly of the cytoskeleton, hint at severing enzymes having broader tasks than only severing microtubules.

Several studies have gained insights into the interaction and coordination of severing enzymes. Using catalytic inactive mutants, it has been shown for spastin and katanin that the assembly of the enzymes into ring-shaped hexamers is needed for severing (Hartman and Vale, 1999; Sandate et al., 2019). Unlike most AAA ATPases, spastin and katanin are monomeric when bound to ADP, and form hexamers in the presence of ATP (Hartman and Vale, 1999; Roll-Mecak and Vale, 2008; Eckert et al., 2012b). The assembly process for hexamer formation is not very well understood; however, for spastin it has been postulated using bulk biochemical assays that the dimerization step, which is promoted by the monomer interaction with microtubules, is crucial in the formation of the active hexameric complex (Eckert et al., 2012a). It is thought that the monomers bind to the microtubules via electrostatic interactions, diffuse along the microtubule surface and oligomerize upon encounter with other monomers. Moreover, the C-terminal tails of tubulin have been found to bind to the central pore of spastin and katanin hexamers (Sandate et al., 2019; White et al., 2007; Zehr et al., 2020). In this configuration, the pull-out of a tubulin subunit from the lattice is supposed to occur due to conformational changes triggered by ATP hydrolysis. Importantly, fluorescence microscopy studies by Vemu et al. (2018) revealed that spastin and katanin can induce nanoscale damage by extracting a few tubulin heterodimers before actual severing occurs. This idea was supported by electron microscopy images and tomograms that visualized severing-mediated defects in which tubulin subunits were missing from the microtubule lattice *in vitro* and *in vivo* (Srayko et al., 2006; Vemu et al., 2018). Interestingly, these damaged sites catalyzed the insertion of GTP-tubulin dimers from solution into the microtubule lattice, leading either to repaired filaments with GTP islands or to rescued stabilized microtubule fragments. Thus, severing enzymes might serve as a quality control system for microtubules by removing dysfunctional tubulin and rejuvenating the microtubule lattice by incorporating fresh GTP-tubulin.

Supporting the latter hypothesis, it was suggested that the severing enzyme katanin (p60/p80 subunits if not otherwise noted) preferentially severs in regions of irregularities in the microtubule lattice: (1) computer modeling of microtubule severing by katanin *in vitro* could only replicate experimental results when lattice defects (Davis et al., 2002) (such as changes in protofilament number; Chretien et al., 1992) were taken into account; and (2) katanin (p60) was observed to localize and sever with increased frequency at the annealing sites between GMPCPP and GDP-taxol

microtubules (Diaz-Valencia et al., 2011). Other studies have demonstrated that katanin has a strong affinity for microtubule crossovers and bundles: (1) in plants, severing by katanin at microtubule crossing sites resulted in alignment and re-orientation of the microtubule cytoskeleton (Lindeboom et al., 2013; Zhang et al., 2013); and (2) in *Caenorhabditis elegans* oocytes, the pruning effect of katanin was confirmed and proposed to contribute to the maintenance of the (anti)parallel spindle architecture (McNally et al., 2014). In addition, katanin was found to be highly substrate selective with respect to tubulin post-translational modifications (PTMs) and microtubule-associated proteins (MAPs), preferentially severing older, more post-translationally modified microtubule parts (Sharma et al., 2007; Benz et al., 2012; Sudo and Baas, 2010). Thus, selective severing might also be a general mechanism for microtubule alignment and rearrangement.

So far, little is known about the mechanistic details of the severing process driven by spastin. Although polyglutamylation, a type of tubulin PTM, has been shown to influence spastin activity (Valenstein and Roll-Mecak, 2016), it is not clear whether spastin has the ability to recognize lattice defects or microtubule intersections. In order to gain deeper insight into the substrate dependence of spastin, we here studied how microtubule defects influence the severing process by spastin *in vitro*. Specifically, we induced defects in the microtubule lattice by either varying the polymerization conditions of microtubules or by end-to-end annealing of microtubules. Subsequently, microtubule severing by spastin was investigated with respect to defect densities and annealing sites. We found that the presence of defects significantly accelerated the onset of severing. Interestingly, our results do not indicate preferential binding of spastin to the defect sites, leading us to propose a model where defects do not actively promote severing but rather passively contribute to microtubule lattice instability.

## RESULTS
### Slow polymerization kinetics results in fewer microtubule lattice defects

It has been shown that the polymerization conditions of microtubules influence the density of defect sites along the microtubule lattice and, hence, their mechanical stability (Schaedel et al., 2019). Therefore, we first investigated whether we could generate two distinct microtubule populations with different defect densities by varying the polymerization parameters. We tested two protocols to polymerize microtubules with the slowly hydrolyzing GTP analog GMPCPP (Fig. 1A, see Materials and Methods). For low defect density, microtubules were polymerized for 5 h at low temperature (28°C) at a moderate tubulin concentration of 2.5 µM (including 25% rhodamine-labeled tubulin). We expected that the resulting slow microtubule assembly would lead to a low defect density. For high defect density, microtubules were polymerized for 30 min at high temperature (37°C) at an elevated tubulin concentration of 20 µM (including 25% Atto647-labeled tubulin). These conditions promote rapid microtubule assembly, which is expected to result in a high defect density.

In order to visualize the lattice structure of the microtubules resulting from the two polymerization protocols, we applied negative stain transmission electron microscopy (TEM) (Fig. 1B). By analyzing the acquired images, we found that the lattices of slowly polymerized microtubules indeed contained significantly fewer visible defects than the rapidly polymerized microtubules (Fig. 1C). The evaluation of the defect types revealed that both microtubules populations exhibited (1) missing pieces or breaks in the microtubule lattice, (2) changes in protofilament number and

(3) open structures due to disrupted lateral interactions (see Fig. 1B for examples of the described defect types). Although missing pieces or breaks in the lattice and open structures were similarly distributed in slowly polymerized microtubules, the number of missing pieces or breaks in the lattice was significantly higher in rapidly polymerized microtubules (Fig. 1D). To ensure that the difference in defect density was due to the polymerization conditions and not because of differently labelled tubulin, we also prepared slowly polymerized microtubules labeled with Atto647 and rapidly polymerized microtubules labeled with rhodamine (dyes exchanged). These microtubules contained similar defect densities and defect type distributions to the microtubules grown under the same conditions with the other dye (Fig. 1C,D; Fig. S1). Thus, microtubules with two different defect densities were successfully grown *in vitro* using different polymerization conditions and are referred to below as microtubules with few or many defects for slow and fast growth, respectively.

### Severing of microtubules with many defects starts earlier than severing of microtubules with few defects

To investigate whether the density of defect sites in the microtubule lattice influences the onset and/or rate of severing by spastin, we immobilized microtubules on a glass coverslip using anti-tubulin antibodies and added spastin at low concentration (≤25 nM). A low spastin concentration allowed us to prolong the severing process (so that differences between individual mechanistic phases could be resolved) as well as to observe localized spastin-microtubule interactions. Because we consistently observed pronounced assay-to-assay variabilities under these conditions (most likely due to a disproportionate influence of inactive spastin molecules on the severing kinetics) we immobilized microtubules with few and many defects side-by-side within the same flow channel (Fig. 2A). This measure ensured identical experimental conditions for microtubules with different defect densities and allowed us to meaningfully compare these populations. Immediately after adding spastin along with 1 mM ATP, we used dual-color TIRF microscopy to follow the severing process of the two microtubule populations (Fig. 2B). During the pre-severing phase, in which spastin binds to microtubules, oligomerizes and starts to pull out tubulin, microtubules stayed intact, before the first severing events were detected at distinct positions in the microtubules. The severing process then proceeded until the resulting microtubule fragments were so small that they detached from the surface and could no longer be reliably followed under the TIRF microscope.

During the observation of the severing, it was already apparent that the timing of the first detected severing events differed for the two microtubule populations. In order to quantitatively compare the severing process for both, we counted the number of severing events that were observed during small time intervals (30 - 60 s) and added them up to plot their cumulative distributions (Fig. 2C–F). The severing onset was determined as the average time of the first 10% of all severing events within one microtubule population. We found that severing started (and finished) earlier for microtubules, with many defects when compared to microtubules with few defects, whereas the severing progress itself was similar and took about the same time from its onset to the completion of severing for both microtubule populations (Fig. 2C). To exclude the possibility that the fluorescent labeling of the tubulin had an influence on the severing process, we performed a number of control measurements. First, we repeated the experiments described above by swapping the fluorophores on the microtubules with many and few defects. We observed again that the microtubules with few defects (now labeled with Atto647) began to sever later and could be observed in the field

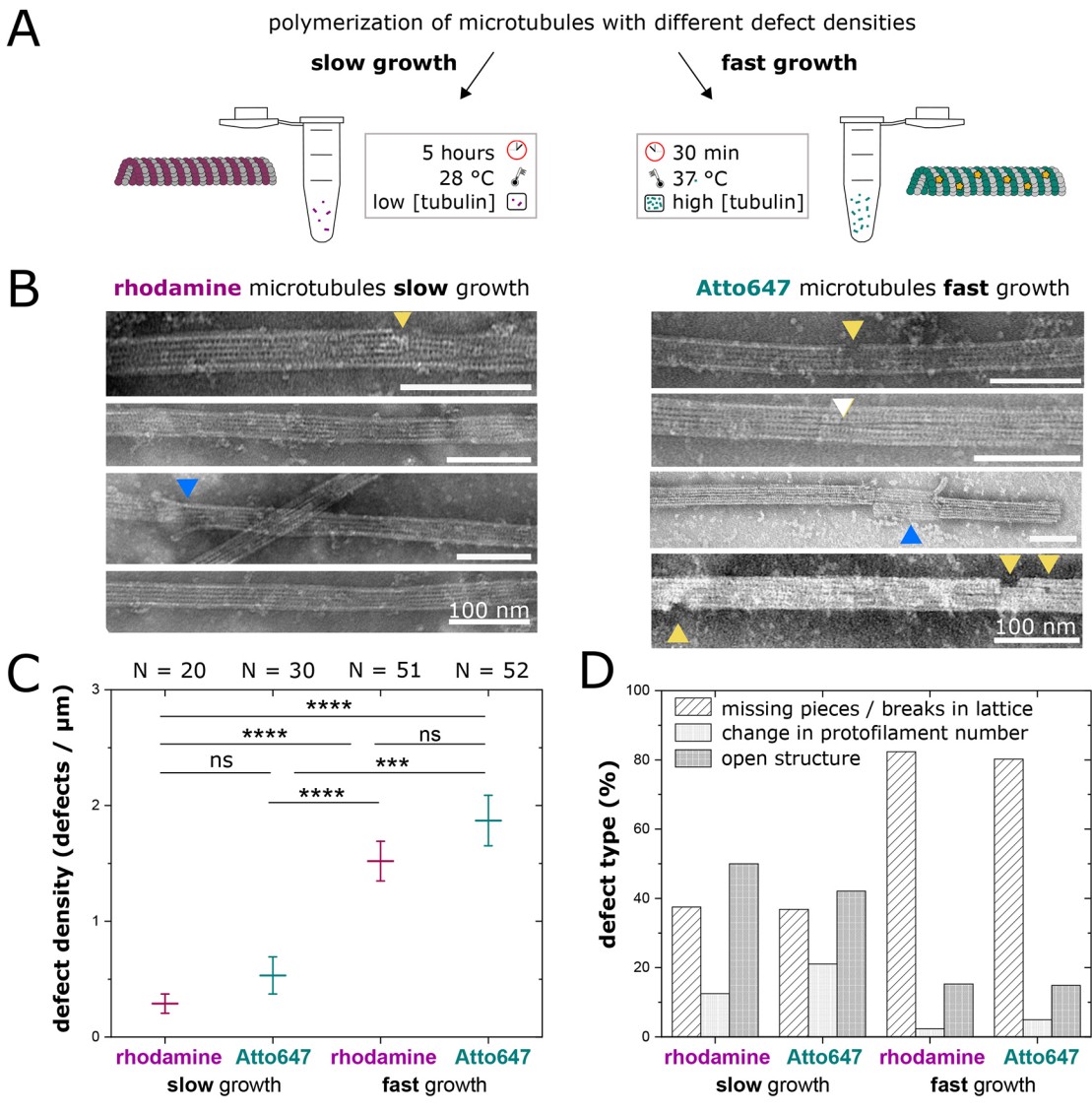

**Fig. 1. Defects in microtubules grown slowly or rapidly.** (A) Schematic representation of two microtubule polymerization protocols used to obtain different defect densities. Magenta indicates rhodamine-labeled tubulin; green indicates Atto647-labeled tubulin; yellow stars indicate lattice defects. (B) Representative transmission electron microscopy images of slowly polymerized microtubules (5 h at 28°C with 2.5 µM rhodamine-labeled tubulin) and rapidly polymerized microtubules (30 min at 37°C with 20 µM Atto647-labeled tubulin). Arrowheads indicate defect sites (yellow indicates a missing piece or break; white indicates a change in protofilament number; blue indicates an open structure). (C) Defect density per µm (mean±s.e.m.) for slowly and rapidly grown microtubules, both labeled with rhodamine and Atto647. *n*=number of microtubules from two independent biological experiments; *P*-values were obtained using a two-tailed Mann–Whitney *U*-test: ****$P<0.0001$, ***$P<0.001$; ns, not significant. (D) Distribution of defect types for slow and fast growth, and two different tubulin dyes in each case.

of view longer than microtubules with many defects (now labeled with rhodamine, Fig. S2A and Movie 1). Likewise, the cumulative severing events over time (Fig. 2D) showed again that severing started and finished earlier for the microtubules with many defects than for the microtubules with few defects, whereas the progression of severing itself remained similar. In each of these experiments, the average severing onset times were significantly different for microtubules with few and many defects, respectively (Fig. S2E). A similar result was also obtained in assays with slightly faster severing kinetics (Fig. S2B,E). As a second control, rhodamine- and Atto647-labeled microtubules, each either with few or many defects, were immobilized together in one flow channel. Upon addition of spastin, both types of microtubules were severed simultaneously, being present for the same amount of time (Fig. S2C,D). No differences were found in the progression of severing (Fig. 2E,F) and the average

severing onset times were not significantly different (Fig. S2E). We therefore conclude that the observed differences in severing between microtubule populations polymerized under different conditions can be attributed to the defect densities.

**Microtubules are preferentially severed at annealing sites**

To investigate whether severing preferentially occurs at defect sites, as compared to other parts of the microtubule, we annealed rhodamine- and Atto647-labeled GMPCPP microtubules with few defects overnight at 30°C (Fig. 3A, steps I and II). End-to-end annealing of microtubules has previously been reported to create microtubule lattice defects at the annealing sites (Diaz-Valencia et al., 2011; Schaedel et al., 2019). Importantly, the dual-color labeling of the microtubules allowed us to localize the annealing sites. To test whether the annealed microtubules were reasonably

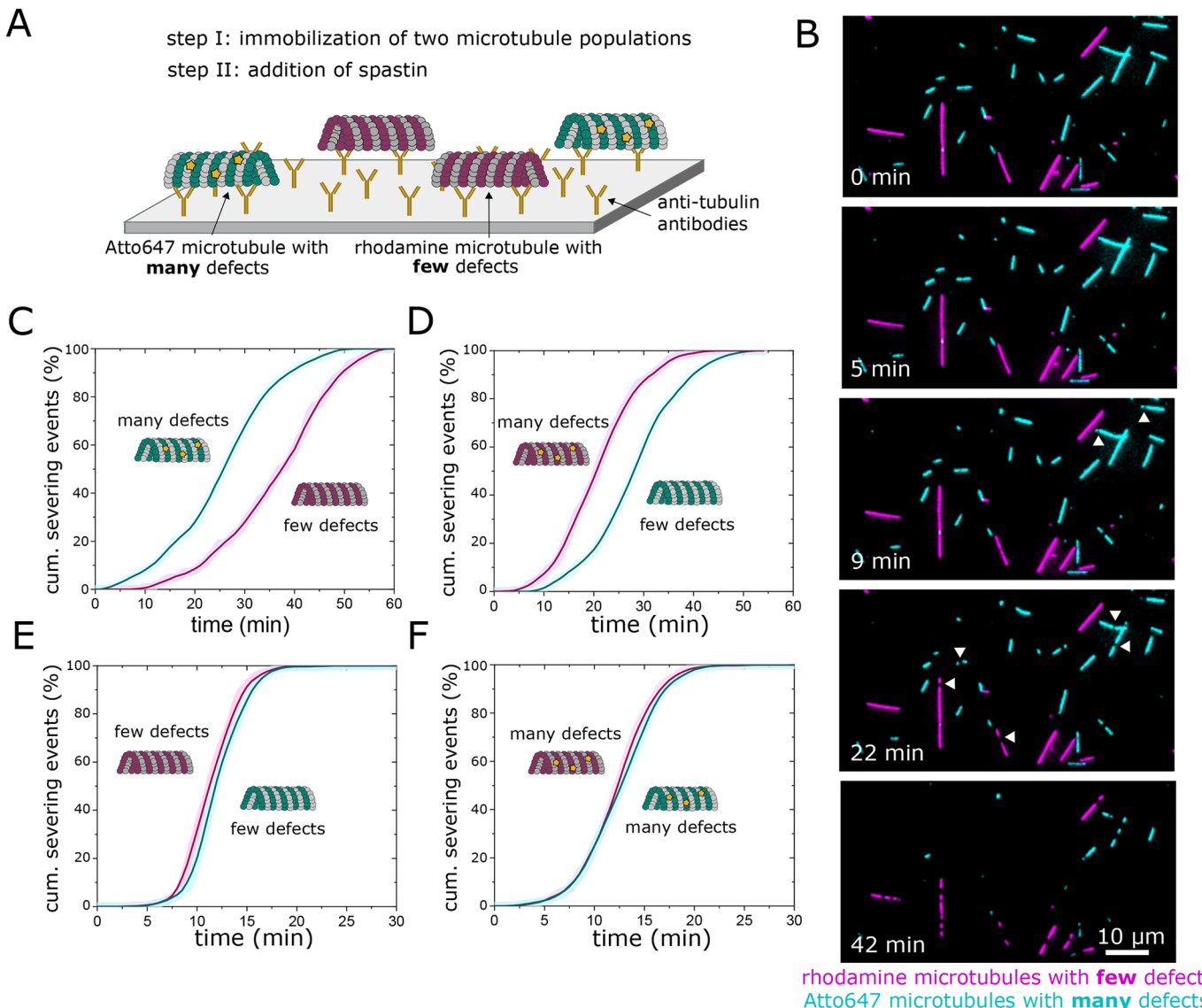

**Fig. 2. Severing time courses of microtubules with different fluorescent tags and defect densities.** (A) Schematic representation of a microtubule-severing assay by spastin. Step I: immobilization of microtubules with few and many defects onto a glass surface coated with anti-tubulin antibodies. Step II: addition of the severing enzyme spastin. (B) Fluorescence micrographs of Atto647 microtubules with many defects being severed faster than rhodamine microtubules with few defects (during a period of 42 min in the presence of 14 nM spastin). White arrowheads indicate the severing sites of single microtubules. (C-F) Cumulative severing events of two microtubule populations as a function of time at spastin concentrations of 14 nM (C,D) and 20 nM (E,F). (C) Rhodamine microtubules with few defects (n=77; total length=470 μm) and Atto647 microtubules with many defects (n=135; total length=483 μm) (corresponding assay in B). (D) Rhodamine (n=125; total length=1029 μm) and Atto647 (n=185; total length=1101 μm) microtubules with few defects. (E) Rhodamine (n=388; total length=1109 μm) and Atto647 (n=297; total length=1054 μm) microtubules with many defects. (F) Rhodamine microtubules with many defects (n=163; total length=531 μm) and Atto647 microtubules with few defects (n=95; total length=573 μm) (corresponding assays are in Fig. S2A,C,D).

stable and did not break apart at the annealing site in the absence of spastin, we performed gliding motility assays on substrate-immobilized kinesin-1 motors (Fig. S3 and Movie 2). We found that the annealed microtubules were gliding smoothly without breaking, indicative of their overall mechanical stability in spite of annealing. Next, we immobilized annealed microtubules on the surface using anti-tubulin antibodies, added 25 nM spastin along with 1 mM ATP and followed the severing process via two-color TIRF microscopy (Fig. 3A, steps III and IV). Severing events could be observed along the lattices of the annealed microtubules at random sites, as well as at the annealing sites (Fig. 3B). To compare the number and timing of severing events between annealing and random lattice sites, we

divided the microtubules into 3 μm regions positioned either around the annealing sites or entirely within the rhodamine-labeled or Atto647-labeled microtubule parts (Fig. 3C). The 3 μm size of these regions was chosen because defects in the microtubule lattice likely not only influence the stability at the annealing sites themselves but also in the surrounding regions (Schaedel et al., 2019). The severing events were quantified by counting them for each region category for all microtubules in a field of view. For the representative microtubule in Fig. 3C, two severing events were observed within the annealed region (region 2, white arrowheads) and one severing event in the non-annealed rhodamine region (region 1, yellow arrowhead). In a population of 68 microtubules, we found a twofold

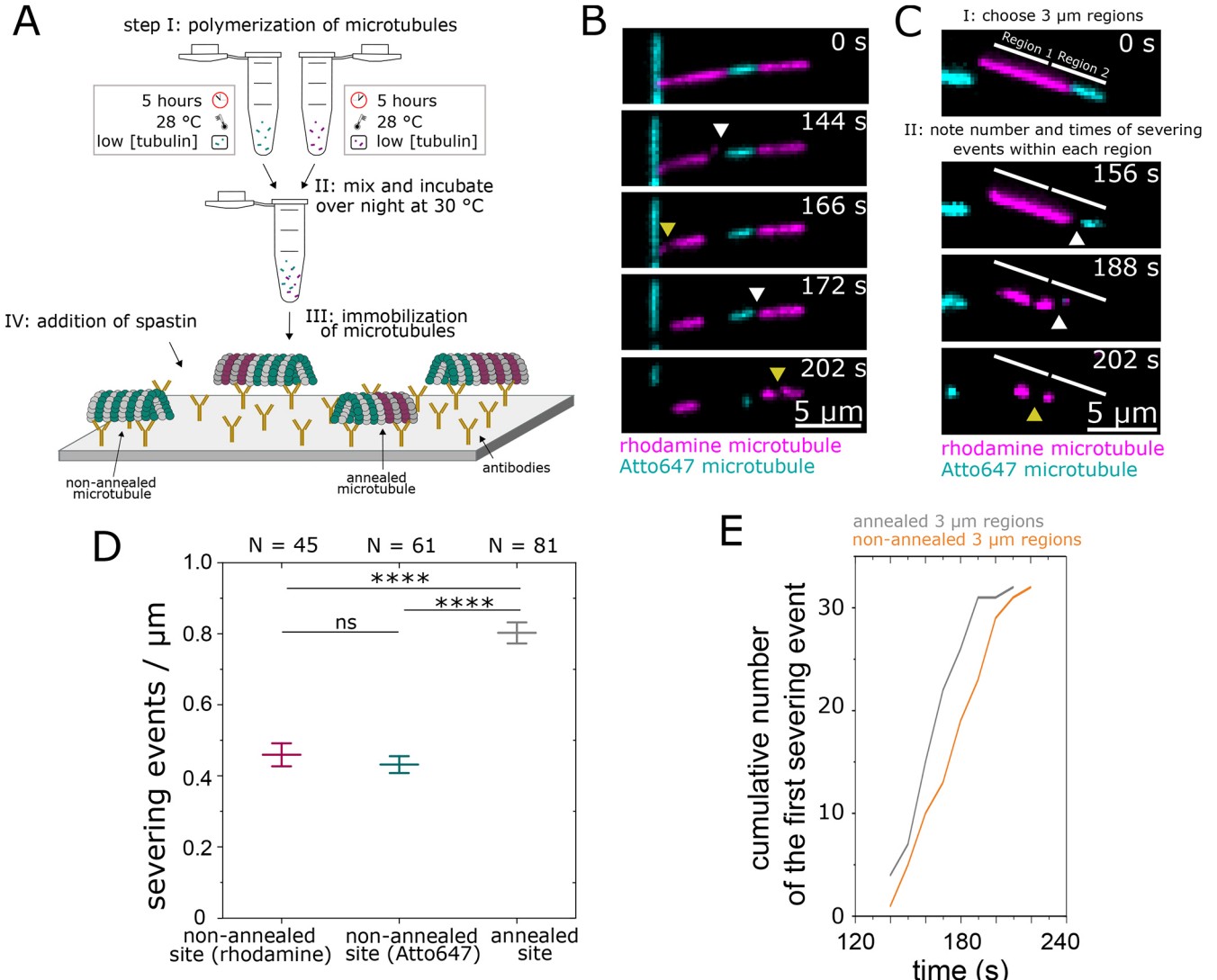

**Fig. 3. Severing of end-to-end annealed microtubules by spastin.** (A) Schematic representation of the steps involved in the severing assay. Step I: polymerization of differently labeled microtubules with few defects. Step II: annealing of microtubules overnight. Step III: immobilization of annealed microtubules onto a glass surface coated with anti-tubulin antibodies. Step IV: addition of the severing enzyme spastin (25 nM). Green indicates Atto647-labeled tubulin; magenta indicates rhodamine-labeled tubulin. (B) Time course of a severing assay, in which a representative annealed microtubule is being severed at its annealing sites (white arrowheads) and at different parts of the microtubule lattice (yellow arrowheads). (C) Analysis steps followed to quantify the severing by spastin at annealing sites compared to non-annealed microtubule lattice sites. Step I: microtubules were divided into 3 µm regions without (region 1: labeled with rhodamine) and with (region 2) an annealing site. Step II: the number and timepoints of the severing events were determined for each region. (D) Number of severing events per µm (mean±s.d.) quantified for rhodamine-only, Atto647-only or annealed 3 µm microtubule regions. $n$=number of microtubules; time period of observation=5 min. $P$-values were obtained using a two-sample unpaired $t$-test: ****$P$<0.0001. (E) Cumulative number of the first severing event within each 3 µm region as a function of time. Equal numbers of annealed and non-annealed regions were compared with each other.

higher number of severing events per µm length of microtubule in regions with annealing sites compared to non-annealed regions with either rhodamine or Atto647 label during the observation period of 5 min (Fig. 3D). No significant differences were observed in the severing events per µm length of microtubule between the rhodamine- and Atto647-only regions. Furthermore, we compared the timepoints of the first severing event within each of the 3 µm regions (Fig. 3E). We found that the severing events in regions around annealing sites occurred at earlier timepoints than in rhodamine- or Atto647-only microtubule regions. These results are consistent with our previous observations, where the onset of microtubule severing occurred earlier for microtubules containing many defects as compared to microtubules containing few defects, independent of the fluorescent tag.

## Tubulin loss and spastin signal evolve equally for microtubules with few and many defects

To examine whether spastin preferentially binds to microtubules with many defects and whether tubulin removal occurs more rapidly, we immobilized Atto647-only microtubules (with many defects) and rhodamine-only microtubules (with few defects) side-by-side in the absence (control) and presence of 7 nM GFP-spastin, and recorded the fluorescence signal of the microtubules and spastin over time (Fig. 4A). In the absence of spastin the total microtubule length [sum of the lengths of all rhodamine-labeled microtubules ($n$=8) at a given time] only decreased slightly over time (less than 0.3% per minute) due to slow inherent microtubule end-depolymerization (Fig. 4B, top panel). In the presence of spastin, the total microtubule length (sum of the lengths of all microtubule

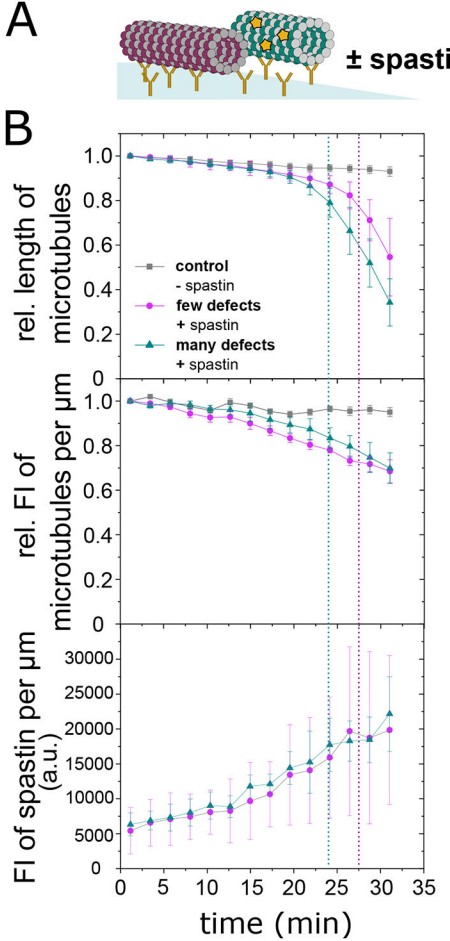

**Fig. 4. Time courses of fluorescence intensity of tubulin and spastin for microtubules with different defect densities.** (A) Schematic of the experimental assay in which rhodamine-labeled microtubules with few defects (magenta) were immobilized next to Atto647-labeled microtubules with many defects (dark cyan) in the absence and presence of 7 nM GFP-labeled spastin. Yellow stars indicate lattice defects. (B) Top: total length of all remaining microtubule parts as a function of time (relative to the value at t=0, mean±s.d., $n$=8 microtubules). Middle: average fluorescence intensity (FI) of microtubules per μm of the total length of the remaining microtubules as a function of time [relative to the value at t=0, mean±s.d.; $P$>0.01 at 8, 10.3, 19.6 and 21.9 min (two-tailed Mann–Whitney $U$-test); $P$>0.05 at all other timepoints]. $n$=7 microtubules (see also Fig. S5 for more data). Bottom: average fluorescence intensity of spastin per μm of the total length of the remaining microtubules as a function of time [mean±s.d.; $P$≥0.086 at all time points (two-tailed Mann–Whitney $U$-test)]. $n$=7 microtubules. Dotted lines indicate the average time of the first severing event per microtubule (magenta indicates microtubules with few defects; cyan indicates microtubules with many defects).

parts remaining at a given time) first decreased slowly during the pre-severing phase (similar to the case without spastin) but accelerated significantly in a non-linear manner after the onset of severing (here determined as the average of the times of the first severing event per microtubule, $t_{so}$). The accelerated decrease in total microtubule length started earlier for microtubules with many defects compared to microtubules with few defects, corresponding to an earlier severing onset at $t_{so}$=23.9±1.0 min (mean±s.e.m., $n$=9 microtubules, cyan dotted line) and a later severing onset at $t_{so}$=27.3±0.7 min (mean±s.e.m., $n$=9 microtubules, magenta dotted line), respectively (significantly different according to a two-tailed Mann–Whitney $U$-test, with $P$=0.024). Similar to the microtubule

length, the fluorescence signal from tubulin decreased faster after the severing onset in microtubules with many defects when the intensity signal was normalized to the initial microtubule length (Fig. S4). However, since we were not primarily interested in this total tubulin loss, but rather in the tubulin loss of the intact-appearing microtubule segments before and after the onset of severing, we normalized the intensity values to the total microtubule length at the respective time point (as defined above). This length-normalized tubulin signal (relative average fluorescence intensity per μm of the total length of the remaining microtubules) decreased only slightly in the absence of spastin (control), likely due to photobleaching (Fig. 4B, middle panel; Fig. S5). In the presence of spastin, the length-normalized tubulin signal decreased significantly faster. Strikingly, this decrease started before the onset of severing (as defined above, see dotted lines), indicating that tubulin removal happens before actual severing is observed. We did not observe a significant difference in the decrease of the length-normalized tubulin signal for microtubules with few and many defects. At the same time, the length-normalized spastin signal (average fluorescence intensity per μm of the total length of the remaining microtubules) increased (Fig. 4B, lower panel; Fig. S6). The persistent increase in the spastin signal over time (i.e. the lack of saturation behavior) likely reflects the oligomerization process of spastin, which is expected to lead to extended dwell times on the microtubule due to cooperative binding and/or lower off-rates after oligomerization. We did not observe a significant difference in time course and intensity of the spastin signals for microtubules with few and many defects. These results suggest that there is neither significantly increased or accelerated binding of spastin to microtubules nor tubulin removal from microtubules with many defects compared to microtubules with few defects.

## DISCUSSION

In order to study the influence of lattice defects on microtubule severing by spastin, we employed two different tubulin polymerization conditions allowing for slow and fast microtubule growth kinetics, resulting in the generation of microtubules with few lattice defects (i.e. low defect density) or many lattice defects (i.e. high defect density), respectively. Using controlled *in vitro* severing assays, we present experimental evidence that a higher defect density in the microtubule lattice results in an earlier onset of the severing process by spastin. Furthermore, we generated localized defect sites by end-to-end annealing of slowly polymerized GMPCPP microtubules. On these microtubules, severing occurred twice as often in the annealing regions after 5 min compared to other regions of the microtubule lattice with presumably fewer defect sites. This result is comparable to earlier findings made with katanin, which has been reported to sever microtubules with ∼50% higher frequency at annealing sites between GMPCPP and Taxol-stabilized GDP microtubules, compared to the regions away from the annealing sites (Diaz-Valencia et al., 2011). In those experiments, katanin was observed to localize quickly to microtubule annealing sites and remained there until severing had taken place. We did not find such defect-site recognition behavior for spastin in our experiments where spastin bound rather uniformly along the microtubule lattice. Even at low spastin concentrations (Fig. S6), we did not observe an obvious accumulation of GFP-spastin at particular sites on the microtubule lattice.

What causes microtubules (or regions of them) with many defects to be severed earlier than microtubules (or regions of them) with few defects? Let us assume the following established mechanism of microtubule severing by spastin: spastin hexamers (which form on

the microtubule lattice) remove tubulin dimers one after another from the microtubule lattice (Kuo and Howard, 2021). Once a certain number of tubulins are removed in close vicinity (i.e. once a given threshold density of missing tubulins in the microtubule lattice is reached locally) the microtubule is severed at this location. In the case of lattice defects, two main mechanisms might be at play. The first is the defect recognition model. Spastin monomers and/or hexamers directly recognize defect sites in the microtubule lattice (i.e. they bind/interact there with higher affinity compared to the rest of the lattice). This leads to a local increase in spastin concentrations, resulting in higher tubulin removal rates at the defect sites. The second is the lattice instability model. Spastin activity on the microtubule lattice is homogeneous along the length, i.e. tubulin removal occurs at random sites. However, at lattice defect sites where tubulins are already missing and the lattice stability is compromised, fewer tubulins need to be removed for severing to occur. For both models, severing would occur earlier on microtubules with many defects compared to microtubules with few defects, because the threshold density of locally missing tubulins is reached faster. Additional synergistic effects are expected for the defect recognition model, because at the defect sites, with missing tubulin subunits, fewer tubulins have to be removed by spastin to reach the critical density required for severing. Moreover, for both models, severing would be expected to occur predominantly in the vicinity of (pre)existing defect sites (either present already at the beginning or generated by spastin itself). Although we cannot make a statement about this correlation for our homogeneously grown microtubules with few and many defects (as we do not have the knowledge about the exact positions of the defect sites), we do know from our experiments with annealed microtubules and from the work of Diaz-Valencia et al. (2011) that severing indeed occurs preferentially near defect sites. Beyond these commonalities, the defect recognition model would predict a higher and rapidly increasing spastin signal, accompanied by a faster decline in the tubulin signal prior to severing, in microtubules with many defects compared to few defects. However, this was not observed in our experiments (Fig. 4B, middle and bottom panels). Instead, our experimental data indicate that the average tubulin and spastin signals are independent of defect density, supporting the lattice instability model.

Taken together, our results suggest that spastin molecules bind randomly to microtubules and remove tubulin subunits without preferential binding or increased affinity for defect sites. Spastin could either create new defects or enlarge existing defects with similar probability. However, in microtubules containing a higher density of defects (particularly those with substantial missing pieces, as frequently observed in fast-grown microtubules, see Fig. 1D) less additional tubulin removal is required to reach the severing threshold, resulting in an earlier onset of severing. The continuous severing progress with similar severing rates for both microtubule types (Fig. 2), even after an earlier onset for microtubules with many defects, indicate that microtubules with few defects behave similarly to those with many, as spastin generates more 'defects' in both. The preference of spastin for specific sites or local accumulations observed *in vivo* (Aiken and Holzbaur, 2024) could be due to factors such as localized microtubule post-translational modifications and MAP decoration (Valenstein and Roll-Mecak, 2016; Lacroix et al., 2010; Tan et al., 2019; Siahaan et al., 2019; Lawrence et al., 2021; Basnet et al., 2018). Some of these factors could 'mark' microtubule lattice defects, leading to preferential severing at these sites – a mechanism also proposed by Kuo and Howard (2021) as an explanation for the

strong affinity of katanin for microtubule crossovers and bundles *in vivo* (Zhang et al., 2013; Lindeboom et al., 2013; Wightman and Turner, 2007; Sharma et al., 2007). Alternatively, the presence of post-translational modifications and MAPs could specifically generate defects at these sites, enabling localized microtubule polymerization and the creation of new microtubule ends (Aiken and Holzbaur, 2024; Vemu et al., 2018; Kuo et al., 2019; Aumeier et al., 2016).

## MATERIALS AND METHODS

### Preparation of flow channels
Experiments were performed in 3 mm wide flow channels composed of a dichlorodimethylsilane (DDS)-coated coverslip (Hyman, 1991) ($22 \times 22$ mm$^2$, Menzel-Gläser) at the bottom, a polyethylene glycol (PEG)-coated coverslip (Papra et al., 2001) ($18 \times 18$ mm$^2$, Menzel-Gläser) on top and two stripes of parafilm as spacers in between by heating the flow sample to 80°C on a hot plate. For gliding motility assays, glass coverslips ($22 \times 22$ mm$^2$, Menzel-Gläser), which were cleaned according to the following procedure, were used at the bottom of the flow channel: sonication in Mucasol/water (1 : 20; v/v) for 15 min was followed by rinsing in deionized water for 2 min. Coverslips were then sonicated in ethanol/water (1:1; v/v) for 10 min, rinsed in deionized water for 2 min, rinsed in MilliQ-water for 2 min and finally dried using a nitrogen airflow.

### Spastin purification
A human spastin plasmid short isoform (Δ227) (Eckert et al., 2012b) was cloned into Optimized Classic Cloning vectors developed by Lemaitre et al. (2019) to produce two different constructs: (1) containing a recombinant N-terminal 6× His-tag and (2) containing a N-terminal 6× His-tag followed by a GFP-tag for fluorescently labeling. The vectors were expressed in *E. coli* cells (BL21, pRARE competent cells). After transformation, cells were pelleted and resuspended in lysis buffer [50 mM sodium phosphate buffer, 300 mM KCl, 5% glycerol, 1 mM MgCl$_2$, 10 mM β-mercaptoethanol and 0.1 mM ATP (pH 7.4)], with 30 mM imidazole and protease inhibitor cocktail (cOmplete, EDTA free, Roche). All the following steps were performed at 4°C. The cells were lysed by flowing the resuspended cell solution through an Avestin Emulsiflex three times. The cell lysate was then centrifuged for 45 min at 40,000 rpm (190,000 *g*) using Type 45 Ti rotor in an Optima XPN-80 ultracentrifuge following the addition of Benzonase (1:10,000, from Sigma Aldrich). The supernatant was filtered through a 0.45 µm membrane and subsequently incubated with Ni-NTA agarose beads (160019892, Qiagen) for 2 h. The beads were washed several times in a 10 ml gravity flow column (Econo-Pacchromatography columns, BioRad) with equilibration buffer (Lysis buffer with 30 mM Imidazole) and eluted with equilibration buffer containing 200 mM imidazole. The 6× His-tag was cleaved over night with PreScission 3C- protease. The uncleaved fraction as well as the His-tagged protease was subsequently removed by incubating with Ni-NTA resin, and the flow-through containing cleaved spastin fraction was concentrated using 10 K MWCo spin filter, snap-frozen in liquid nitrogen and stored at −80°C.

### Kinesin 1 purification
A wild-type kinesin 1 construct consisting of full-length *Drosophila melanogaster* kinesin 1 heavy chain with a C-terminal 6× His-tag was expressed and purified as described previously (Korten et al., 2016).

### Tubulin purification
Porcine tubulin was purified from porcine brain using an established protocols (Castoldi and Popov, 2003). Tubulin was fluorescently labeled with either rhodamine or Atto647N in BRB80 with 5 mM MgCl$_2$, 1 mM GTP and 5% dimethyl sulfoxide (DMSO) at 37°C for 30 min. The ratio of labeled to unlabeled tubulin was 1:3.

### GMPCPP-stabilized microtubules
Two slightly different protocols were used to grow microtubules with few and many defects. Microtubules with many defects were polymerized by

mixing 20 µM rhodamine- or Atto647N-labeled tubulin with 2 mM GMPCPP and 0.1 mM $MgCl_2$ in Brinkley Reassembly Buffer 80 mM [BRB80, adjusted to pH 6.9 with KOH and composed of 80 mM 1,4-piperazinediethanesulfonic acid (PIPES), 1 mM EGTA and 1 mM $MgCl_2$] in a final volume of 10 µl followed by an incubation step of 30 min at 37°C in a dry block heater. Finally, 190 µl BRB80 were added to the microtubule solution that was then centrifuged in an Beckmann Coulter Air-Driven Ultracentrifuge (340401) for 5 min at 150,000 $g$ and resuspended in 150 µl of BRB80. Microtubules with few defects were polymerized with 2.5 µM rhodamine- or Atto647-labeled tubulin, 1.25 mM GMPCPP and 1.25 mM $MgCl_2$ in BRB80 in a final volume of 80 µl for 5 h at 28°C in a dry block heater. Finally, 120 µl BRB80 were added to the microtubule solution that was then centrifuged in a tabletop centrifuge at 17,000 $g$ for 15 min and resuspended in 150 µl of BRB80.

### Preparation of end-to-end annealed microtubules
In order to anneal differently labeled GMPCPP microtubules, rhodamine and Atto647-labeled microtubules were polymerized separately by following the protocol for microtubules with few defects. Equal volumes of rhodamine and Atto647 microtubules were then mixed and incubated overnight at 30°C. Before starting the experiment, the microtubule mixture was pelleted in a tabletop centrifuge at 17,000 $g$ for 15 min and resuspended in 150 µl of BRB80.

### Microtubule severing assays
First, flow cells were flushed with 20 µl of the following sequence of solutions to immobilize microtubules on the surface: (1) a 1500× dilution of TetraSpeck microspheres (0.1 µm diameter, 1786280 Thermo Fisher Scientific) to allow drift correction; incubation time 5 min; (2) a solution of monoclonal anti-β-tubulin I mouse antibodies; incubation time 5 min; (3) two flushes of BRB80 to wash the flow cell; (4) BRB80C buffer (0.45 mg/ml Casein in BRB80) for blocking the surface from unspecific protein adsorption; incubation time 5 min; (5) a 10× dilution of the GMPCPP-stabilized microtubules in BRB80; incubation time 10 min; and (6) antifade solution (110 µg/ml glucose oxidase and 22 µg/ml catalase in HEPES-based buffer). For severing the immobilized microtubules, 24 µM spastin or 34.3 µM GFP-spastin was first diluted 10–40× in HEPES buffer (20 mM HEPES, 75 mM KCl, 0.5 mg/ml casein, 0.1% Tween-20, 1 mM ATP, 20 mM glucose and 10 mM DTT) and then further in antifade solution in HEPES buffer to achieve the final concentration indicated for each experiment. The spastin dilution was then perfused into the flow cell, which was already placed on the microscope, and imaging was started immediately.

### Gliding motility gliding assays
The flow cell was first perfused with 20 µl of BRB80C buffer to coat the cleaned coverslip with casein. The solution was left to adsorb for 5 min. 20 µl of a kinesin 1 solution (2 µg/ml kin-1, 10 mM DTT, 1 mM ATP and 0.2 mg/ml Casein in BRB80) were then added and incubated for another 5 min. Thereafter, 20 µl of motility solution (1 mM ATP, 20 mM glucose, 10 mM DTT, 33 nM microtubules, 56 µg/ml glucose oxidase and 11 µg/ml catalase in BRB80) containing microtubules was applied. After 5 min, unbound microtubules were washed out with antifade solution in BRB80 (1 mM ATP, 20 mM glucose, 10 mM DTT, 20 µg/ml glucose oxidase and 10 µg/ml catalase in BRB80) and the gliding microtubules were imaged.

### Optical microscopy
All the experiments were performed at 28°C using an objective heater unless indicated otherwise. An inverted fluorescence microscope (Axio Observer Z1) with a 63× oil immersion 1.46 NA objective (Zeiss) and an EMCCD camera (Andor Technology) controlled by the software Metamorph 7.7.7 was used. A built-in magnification (1.6× Optovar) was additionally implemented, leading to a pixel size of 0.159 µm. Rhodamine- or Atto647-labeled GMPCPP-stabilized microtubules were imaged via epifluorescence or TIRF microscopy, indicated for each experiment. For epifluorescence illumination, a Sola SE2 (13212) Lumen 200 metal arc lamp with a GFP, TRITC or Cy5 filter was used. TIRF imaging was performed by using the Omicron Laserbox (Rodgau-Dudenhofen, with the 488 nm, 561 nm and 642 nm lasers) and a QuadLine TIRF filter (AHF, 405/488/561/640×) and

adjusting the TIRF angle before starting with an experiment. A fast filter wheel (Finger Lakes Instrumentation) was employed to image two colors nearly simultaneously (time difference=123 ms) in some of the experiments, as indicated. An exposure time of 100 ms was employed. The time between frames was 2 s in experiments in which the severing of microtubule with few and many defects was investigated over time. The fluorescence signal of GFP-spastin on microtubules was acquired in stream mode (10 images/s), and about every 130 s an image of the microtubules was taken. To determine the change in microtubule fluorescence intensity over time (Fig. S5), imaging was performed using a Nikon Eclipse Ti2 microscope equipped with a perfect focus system and a 100×1.49 NA oil, apochromat TIRF objective and 1.5×optovar. Samples were illuminated with 550 nm (75%) and 660 nm (25%) of a CoolLED pE-4000 lamp. Images from different fluorescent channels were acquired with separate EMCCD cameras (iXon Life EMCCD for 550 nm and iXon Ultra EMCCD for 660 nm), each containing 1024×1024 pixel sensor and controlled with VisiView. The size of each pixel was 87×87 nm. Images were acquired in time-lapse mode with 100 ms exposure (1 frames every 5 s). Apart from photobleaching, for which we have performed controls, we do not expect any other optical effects that could alter our results in Fig. 4 and Fig. S5: If fluorophore quenching were to play a role, we would expect an increase in the tubulin signal upon tubulin removal by dequenching and a decrease in the spastin signal due to quenching. Both effects would tend to reinforce the trends we observed.

### Transmission electron microscopy
Carbon-coated 400 mesh copper transmission electron microscopy (TEM) grids (S160-4 Plano, Wetzlar) were treated with plasma etching for 15 s to make the surface of the grid hydrophilic and enable the adhesion of microtubules. After centrifugation, the pellet of microtubules was then resuspended in 100 µl BRB80. 5 µl of the resuspended microtubule solution were carefully placed on the grid and incubated for 10 min. The excess sample solution was absorbed using a clean paper towel from the edges of the grid. The grids were allowed to dry completely for 20 min. To subsequently stain the microtubules, the dried grids were washed with 10 µl of ultrapure water and then 10 µl of the staining solution (0.75% uranyl formate solution containing 25 mM sodium hydroxide solution) was applied for 15 s. Finally, the staining solution was carefully removed with a clean paper towel and the grids were allowed to dry overnight. The TEM samples were imaged using a FEI Morgagni 268D transmission electron microscope operated at 80 kV in combination with an Olympus MegaView III camera of the CMCB Electron Microscopy Facility located at the Center for Regenerative Therapies Dresden.

### Data processing and analysis
Severing events (microtubule breaks) were counted manually as a function of time, from the beginning of the timelapse movie immediately after the addition of spastin until no more microtubules were in the field of view (i.e. when no further severing events were possible). Severing events were plotted as cumulative events normalized from 0 to 100. To quantify the number of severing events per µm microtubule length in regions around an annealing site compared to a random microtubule lattice site (labeled with either rhodamine or Atto647), we divided each microtubule into 3 µm regions, as shown in Fig. 3C for a representative microtubule. For each type of region (rhodamine only, Atto647 only or annealed), the number and timing of severing events was determined. ImageJ software was employed for image processing. Fluorescence intensity data were acquired by tracking microtubules with Fluorescence Image Evaluation Software for Tracking and Analysis (FIESTA) (Ruhnow et al., 2011) (Fig. S5), which automates Gaussian fitting of fluorescence signals (or in Fig. S6 by line scans along microtubules using ImageJ software). Tracked data were manually corrected to exclude erroneous tracks. Since microtubule intensity data was only collected~every 130 s, the average of the two time points before and after each stream of GFP-spastin was plotted at the same time point as the corresponding spastin signal.

### Acknowledgements
We thank Foram Joshi and the Electron Microscopy Facilities at the Center for Molecular and Cellular Bioengineering (CMCB) of TUD Dresden University of

Technology for expert support with electron microscopy, Günther Woehlke for the kind donation of the spastin plasmid, Corina Bräuer for technical assistance, and all members of the Diez laboratory for scientific discussions.

**Competing interests**
The authors declare no competing or financial interests.

**Author contributions**
Conceptualization: C.R., P. S.-O., S.D.; Investigation: C.R., P. S.-O.; Formal analysis: C. R., P.S.-O.; Resources: R. G., T. K.; Supervision: S.D.; Writing – original draft: C.R., P.S.-O., S. D.; Writing – review & editing: C. R., P. S.-O., R.G., T.K., S.D.

**Funding**
This work was funded by the TUD Dresden University of Technology. Open Access funding provided by TUD Dresden University of Technology. Deposited in PMC for immediate release.

**Data and resource availability**
All relevant data and details of resources can be found within the article and its supplementary information.

**Peer review history**
The peer review history is available online at https://journals.biologists.com/jcs/lookup/doi/10.1242/jcs.264497.reviewer-comments.pdf

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
