## [Peer Review File · Journal of Cell Science]

Microtubule lattice defects facilitate spastin-mediated severing

Cordula Reuther, Paula-Santos Otte, Rahul Grover, Till Korten and Stefan Diez

DOI: 10.1242/jcs.264497

Editor: Michael Way

Review timeline

Original submission:	7 October 2025
Editorial decision:	10 November 2025
First revision received:	11 February 2026
Accepted:	11 February 2026

Original submission

First decision letter

MS ID#: jcs.264497

MS TITLE: Microtubule lattice defects facilitate spastin-mediated severing

AUTHORS: Cordula Reuther; Paula Santos Otte; Rahul Grover; Till Korten; Stefan Diez

ARTICLE TYPE: Research Article

Dear Stefan,

We have now reached a decision on the above manuscript.

As you will see, the reviewers are positive raise a number of questions / criticisms that prevent me from accepting the paper at this stage. They suggest, however, that a revised version might prove acceptable, if you can address their concerns. If you think that you can deal satisfactorily with the criticisms on revision, I would be pleased to see a revised manuscript.

Reviewer 1

Advance summary and potential significance to field

The manuscript by Reuther et al. investigates whether lattice defect density influences spastin's microtubule-severing dynamics. Overall, the experiments are conceptually well designed, with clear distinction between two polymerization regimes, use of GMPCPP to stabilize the lattice, TEM imaging for structural validation, and appropriate dye-swap controls. The data convincingly demonstrate that spastin preferentially severs defect-rich microtubules. The observation that tubulin signal loss precedes severing is potentially significant, as it suggests a pre-severing lattice destabilization mechanism. However, the study would benefit from stronger statistical support, a clearer definition and quantification of defect density, and a more detailed mechanistic discussion. Enhancing the quantitative analysis and clarifying the temporal relationships among tubulin loss, spastin accumulation, and severing events would substantially strengthen the manuscript's conclusions.

Comments for the author

Minor comments

- 1) The operational definition of "many" versus "few" defects is unclear. How were these defect densities determined or quantified? Was this based on prior imaging, labeling conditions, or physical characterization (e.g., protofilament loss or lattice irregularities)? This is essential to interpret the differences between Atto647 and rhodamine microtubules.
- 2) TEM drying/fixation can introduce breaks or open structures. Run a subset of samples by cryo-EM (vitreous freezing) or at minimum negative-stain EM to confirm defects are intrinsic, not prep-induced.
- 4) For figure 1B, I would suggest authors to provide p-values or confidence intervals for the defect-density difference. Report the number of microtubules and independent biological replicates more clearly.
- 5) To exclude the possibility that the fluorescent labeling of the tubulin had an influence on the severing process, authors swapped dyes, excellent idea. Also show a control with unlabeled tubulin (0% dye) to confirm dye incorporation is not subtly changing polymerization or lattice stability. Quantify labeling efficiency (mol dye per tubulin) and ensure equal across conditions.
- 6) For Figure 1C, the difference in average onset times (5.1 vs. 13.4 min; 6.3 vs. 11.0 min) suggests a strong effect of defect density, but statistical analyses (e.g., n values, standard deviation, significance tests) are missing. Please report the number of microtubules analyzed, measures of variability, and p-values to substantiate the conclusions.
- 7) The described "lag phase" before severing onset could reflect spastin binding kinetics, lattice remodeling, or ATP-dependent activation. Some discussion or modeling of this phase would improve mechanistic insight, particularly regarding whether defect density influences spastin binding vs. severing efficiency.
- 9) The main assay uses 14 nM spastin, but a later control mentions 20 nM and 25 nM. It would be helpful to comment on whether spastin concentration affects the observed kinetics or whether the defect-dependent effect persists across a range of enzyme concentrations.
- 10) For Figure 4, the text provides average onset times (24.0 min vs 27.5 min) for severing but does not report measures of variability (e.g., standard deviation, SEM, or confidence intervals) or statistical tests to determine significance.
- 11) For Figure 4, the number of microtubules analyzed (N = 9 per condition) may be low for robust statistical analysis.
- 12) The conclusion that tubulin removal occurs before severing is intriguing but requires stronger evidence. Could this pre-severing decrease reflect partial bleaching, fluorophore quenching, or structural rearrangements rather than subunit removal? It would help to include controls for illumination intensity, exposure duration, and normalization to photobleaching rates.
- 13) The study aims to test "dependence on defect density," but the results conclude that spastin binding and signal increase are not significantly different between high- and low-defect microtubules. The authors might clarify whether the difference in severing onset is driven by local structural accessibility rather than binding kinetics – or discuss alternative interpretations.

Reviewer 2

Advance summary and potential significance to field

The manuscript titled "Microtubule lattice defects facilitate spastin mediated severing" provides strong experimental evidence that spastin's microtubule severing activity is influenced by lattice defect density, supporting the lattice instability model over the defect recognition model. The authors convincingly performed in vitro assays using microtubules with few or many defects and demonstrated that microtubules with higher defect densities exhibit an earlier onset of severing. They provided experimental evidence that this early severing is due to reduced number of tubulins that need to be removed at defect sites for microtubule severing. Further, their severing experiment using annealed rhodamine- and Atto647-labeled GMPCPP microtubules with few defects and the observation that severing predominantly occurs near annealed sites strengthen their conclusions. Additionally, uniform spastin-GFP binding to the microtubule lattice further supports the lattice instability model.

Overall, the paper is well written and data are presented in a clear manner. The experiments are conducted carefully and include appropriate controls such as reciprocal polymerization experiments

with the two labeled tubulins and gliding assays to verify the integrity of annealed microtubules. Nonetheless, I have a few suggestions to further improve this article:

P5, Line #18-21 - The authors observed deformities in microtubules during both rapid and slow polymerization. It would be beneficial to know whether they noted a predominance of any specific defect type, such as missing or broken microtubule pieces in the lattice, changes in protofilament number, or open structure. Additionally, it would be informative to determine whether these defect types were uniformly distributed across the two polymerization methods or whether there was a bias toward certain defects during rapid polymerization.

P7- In the control experiments, where the authors have swapped the fluorophores, there is a considerable difference in the severing kinetics in the first 30 minutes of the experiment. The length of microtubules with many defects that remained after 30 min was 37%, whereas it was only 19% in the fluorophore-swapping experiment. Similarly, it was 70% and 50 % in the case of MT with fewer defects. There is no explanation for these differences in the text.

P8, In figure 2, there is no difference in the severing rate between microtubules with few and many defects at 20 nM spastin. If anything, the microtubules with few defects seem to get cut faster. This is puzzling and needs explanation.

P12, Since microtubule severing starts and finishes sooner for ones with more defects, I would have expected a greater decrease in tubulin signal for those microtubules in figure 4b, which is not observed. Specifically, I would have expected a difference between the cyan and magenta curves in the middle panel (rel. Fl of microtubules per μm) after 25 min. Unless I am misunderstanding how this analysis was performed, this discrepancy requires an explanation.

P16, Line#20 - Provide the make of "airfuge"

P18, Line#28 - correction needed, perhaps revise to: "Finally, the grids were stained with a 0.75 % uranyl formate solution...."

First revision

Author response to reviewers' comments

We thank the reviewers for their careful and constructive review of our manuscript. Please find below our detailed point-to-point responses to the reviewers' comments and our actions taken. In addition, the key changes are also indicated by blue color in a submitted version of the manuscript for review only.

In particular, we now (i) present additional data from the evaluation of the TEM-images (defect density and defect types for each combination of dye-polymerization condition) including statistical analyses, (ii) explain the experimental design due to the variability of the assays in more detail, (iii) present Figure 1 and 2 in a restructured manner, (iv) clarify the term 'lag phase' and the determination of the severing onset times, (v) present recalculated average severing onset times including statistical analyses, (vi) clarify the analysis of the tubulin signal in Figure 4B and (vii) address concerns regarding low sample size and pre-severing tubulin loss Figure presentations and legends have been revised accordingly.

Comments from the Reviewers:

Reviewer 1: SUMMARY OF THE ADVANCE MADE IN THIS PAPER AND ITS POTENTIAL SIGNIFICANCE TO THE FIELD

The manuscript by Reuther et al. investigates whether lattice defect density influences spastin's

microtubule-severing dynamics. Overall, the experiments are conceptually well designed, with clear distinction between two polymerization regimes, use of GMPCPP to stabilize the lattice, TEM imaging for structural validation, and appropriate dye-swap controls. The data convincingly demonstrate that spastin preferentially severs defect-rich microtubules. The observation that tubulin signal loss precedes severing is potentially significant, as it suggests a pre-severing lattice destabilization mechanism. However, the study would benefit from stronger statistical support, a clearer definition and quantification of defect density, and a more detailed mechanistic discussion. Enhancing the quantitative analysis and clarifying the temporal relationships among tubulin loss, spastin accumulation, and severing events would substantially strengthen the manuscript's conclusions.

SUGGESTIONS TO AUTHORS

Minor comments (Note: there were no comments 3 or 8 provided by the reviewer)

1) The operational definition of "many" versus "few" defects is unclear. How were these defect densities determined or quantified? Was this based on prior imaging, labeling conditions, or physical characterization (e.g., protofilament loss or lattice irregularities)? This is essential to interpret the differences between Atto647 and rhodamine microtubules.

Reply: We apologize that the quantification of the defect densities was not clear. Our definition of "many" and "few" defects is based on the analysis of defect density by counting the visible defects in the microtubule lattices in the electron microscope images. Although we acquired data for four different combinations of dye-polymerization condition, the defect densities reported in the manuscript were classified and averaged only according to the polymerization condition. **We now state the defect density for each combination of dye-polymerization condition (added Figure 1C). In addition, we have determined the defect types of the counted irregularities and divided them into three categories: (i) missing pieces / breaks in lattice, (ii) changes in protofilament number, and (iii) open structures (added Figure 1D; see also Reviewer 2, first question).**

2) TEM drying/fixation can introduce breaks or open structures. Run a subset of samples by cryo-EM (vitreous freezing) or at minimum negative-stain EM to confirm defects are intrinsic, not prep-induced.

Reply: Negative stain electron microscopy is a common method for imaging microtubules or other biological molecules involving a standard transmission electron microscope (TEM). In our work we used uranyl formate (see methods) to negatively stain/fix the microtubules on a copper grid. We are aware that images from negative-stain EM are not artefact free, that resolution is limited by the granularity of the stain, and that microtubules may appear flattened or suffer from dehydration. However, we observed clear differences in the defect densities of the two microtubule populations, which were polymerized at different conditions but treated identically prior to imaging. Therefore, the differences cannot be explained by the sample preparation method. We believe that our results from negative stain EM are sufficient, in the scope of our work, as we were looking for relative differences in the microtubule preparations and **now also show in more detail that this is independent of the dyes (see also answer to comments 1 and 4).** Further study and classification of the nature of the defect types by cryo-EM might be of interest for future studies.

4) For figure 1B, I would suggest authors to provide p-values or confidence intervals for the defect-density difference. Report the number of microtubules and independent biological replicates more clearly.

Reply: We now provide p-values for the defect-density difference (for each combination of dye-polymerization condition) and report number of microtubules and independent biological replicates more clearly (Figure 1C).

5) To exclude the possibility that the fluorescent labeling of the tubulin had an influence on the severing process, authors swapped dyes, excellent idea. Also show a control with unlabeled tubulin (0% dye) to confirm dye incorporation is not subtly changing polymerization or lattice stability. Quantify labeling efficiency (mol dye per tubulin) and ensure equal across conditions.

Reply: Whether the fluorescent dye influences the results in *in vitro* experiments is always an important question. Since we cannot exclude the possibility that the incorporation of the dye subtly alters polymerization and lattice stability, and that fluorescence-based imaging may also introduce defects/breaks in the microtubule due to photodamage, it was essential for our experiments to compare microtubules with equal prerequisites (i.e. both labeled with a fluorescent dye) and using the same imaging method. This means that both types of microtubules (with few and many defects) were fluorescently labeled and imaged using fluorescence microscopy. To rule out the possibility that one dyes makes more defects than the other, we swapped the dyes and obtained similar results. If dyes generally make microtubules more or less stable would not change our results. This could potentially be tested in future experiments by polymerizing labeled and unlabeled microtubules under the same conditions, flowing them into the flow cell one after the other (so that they can be distinguished later), and imaging with IRM (interference reflection microscopy).

6) For Figure 1C, the difference in average onset times (5.1 vs. 13.4 min; 6.3 vs. 11.0 min) suggests a strong effect of defect density, but statistical analyses (e.g., n values, standard deviation, significance tests) are missing. Please report the number of microtubules analyzed, measures of variability, and p-values to substantiate the conclusions.

Reply: We agree that the differences in average onset times were not well characterized and explained. We now explain the variability of the assays in more detail (see also Reviewer 2, question 2), have restructured Figure 1 and 2, have defined the severing onset time more clearly and performed statistical analyses (Figure S2E).

7) The described "lag phase" before severing onset could reflect spastin binding kinetics, lattice remodeling, or ATP-dependent activation. Some discussion or modeling of this phase would improve mechanistic insight, particularly regarding whether defect density influences spastin binding vs. severing efficiency.

Reply: In the literature the pre-severing or "lag" phase was described as the the time from the start of the assay (i.e. the addition of spastin and ATP to the flow cell) to the time point at which microtubule severing occurred/started (Eckert et al., 2012). It is the phase in which spastin molecules bind to the microtubule, diffuse, hexamerize and pull out tubulin until an actual severing event can be observed microscopically (described in the main text). The length of this phase depends on the spastin concentration and the amount of inactive molecules. It can be further influenced by other parameters, e.g. free tubulin in solution, temperature, level of tubulin glutamylation, etc. (Kuo et al., 2019, Reuther et al. 2022, Valenstein and Roll-Mecak, 2016). From our point of view, neither spastin binding nor tubulin removal (Fig. 4B) is influenced by the lattice defect density but severing efficiency (i.e. the timing of observed severing) is. **We now introduce and discuss the term pre-severing phase in the main text.**

9) The main assay uses 14 nM spastin, but a later control mentions 20 nM and 25 nM. It would be helpful to comment on whether spastin concentration affects the observed kinetics or whether the defect-dependent effect persists across a range of enzyme concentrations.

Reply: The defect-dependent effect on the severing onset times persists over a range of spastin concentrations. However, at low spastin concentrations, such as those used in the experiments presented, the severing process was prolonged, allowing differences between individual mechanistic phases to be better resolved. **We now explain the experimental design in more detail and show another cumulative severing curve (with faster severing kinetics than in Fig. 2 C and D) from an assay with microtubules with few and many defects.**

10) For Figure 4, the text provides average onset times (24.0 min vs 27.5 min) for severing but does not report measures of variability (e.g., standard deviation, SEM, or confidence intervals) or statistical tests to determine significance.

Reply: We thank the reviewer for pointing out this missing statistical analysis. **We updated the manuscript with the recalculated average severing onset times including the s.e.m. and Mann-Whitney U-test for determining significance.**

11) For Figure 4, the number of microtubules analyzed (N = 9 per condition) may be low for robust

statistical analysis.

Reply: We agree that the $N = 9$ microtubules per condition is rather low. However, we needed to have them all in one field of view and we had to exclude microtubules that were crossing each other. In order to increase the number of microtubules we would have to add data from another experiment, but this would then have not necessarily the same data due to the assay variability at low spastin concentrations (see our reply to reviewer 2, question 2). **We repeated this and similar experiments and show part of this data in Fig. S5C and D ($N = 12$ and 14), yielding the same trends. We are thus confident that the data displayed in Figure 4 is representative and have applied an appropriate statistical test.**

12) The conclusion that tubulin removal occurs before severing is intriguing but requires stronger evidence. Could this pre-severing decrease reflect partial bleaching, fluorophore quenching, or structural rearrangements rather than subunit removal? It would help to include controls for illumination intensity, exposure duration, and normalization to photobleaching rates.

Reply: We agree that it is important for this analysis to exclude all effects caused by illumination. Therefore, we included bleaching controls in Fig. 4 and Fig. S5B, C, D and further controls (e.g. by the presence of spastin without ATP) in Fig. S5A. In these individual control experiments, control data were acquired under exactly the same imaging conditions (illumination intensity, exposure rate, imaging rate) as the data with spastin. However, in-between different experiments imaging conditions often varied (compare Fig. 4 to Fig. S4). Nevertheless, as all these experiments showed similar results, we believe that illumination intensity and exposure duration do not affect our results and their interpretation significantly. If fluorophore quenching would contribute to our findings, we would expect an increase in the tubulin signal upon tubulin removal via dequenching and a decrease in the spastin signal due to quenching. Both effects would rather enhance our observed trends. Moreover, we do not expect a change in fluorescence intensities due to structural rearrangements. **We now added these reasonings into the materials and methods section and the Figure caption of Fig. S5.**

13) The study aims to test "dependence on defect density," but the results conclude that spastin binding and signal increase are not significantly different between high- and low-defect microtubules. The authors might clarify whether the difference in severing onset is driven by local structural accessibility rather than binding kinetics – or discuss alternative interpretations.

Reply: In our study, we set out to test how microtubule lattice defects influence spastin-mediated severing. While we observe no significant difference in the spastin binding kinetics between few and many defect microtubules, we find that the onset of severing clearly depends on defect density.

Here is our reasoning: The severing onset reflects the combined processes of spastin binding, tubulin removal, and progression toward a lattice stability threshold at which breakage occurs. Because neither spastin binding (Fig. 4B, lower panel) nor tubulin removal rates (Fig. 4B, middle level) differ between conditions, we conclude that the earlier severing onset observed in highly defective microtubules is most consistently explained by reduced lattice stability, such that the critical stability threshold is reached more rapidly.

If local structural accessibility or preferential hexamerization at defect sites were the dominant factor, one would expect increased spastin binding to high-defect microtubules overall or over time, which we do not observe (Fig. 4C). Based on these findings, we therefore propose a lattice instability model to account for the defect-dependent differences in severing onset. **We checked and slightly modified our text to assure that this reasoning is comprehensible for the reader.**

Reviewer 2: The manuscript titled "Microtubule lattice defects facilitate spastin mediated severing" provides strong experimental evidence that spastin's microtubule severing activity is influenced by lattice defect density, supporting the lattice instability model over the defect recognition model. The authors convincingly performed in vitro assays using microtubules with few or many defects and demonstrated that microtubules with higher defect densities exhibit an earlier onset of severing. They provided experimental evidence that this early severing is due to reduced number of

tubulins that need to be removed at defect sites for microtubule severing. Further, their severing experiment using annealed rhodamine- and Atto647-labeled GMPCPP microtubules with few defects and the observation that severing predominantly occurs near annealed sites strengthen their conclusions. Additionally, uniform spastin-GFP binding to the microtubule lattice further supports the lattice instability model.

Overall, the paper is well written and data are presented in a clear manner. The experiments are conducted carefully and include appropriate controls such as reciprocal polymerization experiments with the two labeled tubulins and gliding assays to verify the integrity of annealed microtubules. Nonetheless, I have a few suggestions to further improve this article:

P5, Line #18-21 - The authors observed deformities in microtubules during both rapid and slow polymerization. It would be beneficial to know whether they noted a predominance of any specific defect type, such as missing or broken microtubule pieces in the lattice, changes in protofilament number, or open structure. Additionally, it would be informative to determine whether these defect types were uniformly distributed across the two polymerization methods or whether there was a bias toward certain defects during rapid polymerization.

Reply: We thank the reviewer for this proposal. We now analyzed the data with respect to defect types and divided them into three groups: missing or broken microtubule pieces in the lattice, changes in protofilament number, or open structure. While open structures and missing pieces/breaks in the lattice occurred with approximately similar frequency during slow growth, missing pieces/breaks in the lattice were the most common defect type during fast growth. The dye had no influence on the distribution of defect types. **The data is now also displayed and reported in the manuscript (Fig. 1D).**

P7- In the control experiments, where the authors have swapped the fluorophores, there is a considerable difference in the severing kinetics in the first 30 minutes of the experiment. The length of microtubules with many defects that remained after 30 min was 37%, whereas it was only 19% in the fluorophore-swapping experiment. Similarly, it was 70% and 50 % in the case of MT with fewer defects. There is no explanation for these differences in the text.

Reply: We apologize for the confusion caused by the lack of explanation in the original text. The differences noted by the reviewer primarily arise from assay-to-assay variability, which we consistently observe when working at very low spastin concentrations. At such low concentrations, variability is amplified compared to studies using substantially higher spastin levels (e.g., Eckert et al. 2012; Valenstein and Roll-Mecak 2016; Kuo et al. 2019; Reuther et al. 2022). Under these conditions, inactive spastin - which is inevitably present in purified protein preparations - likely has a disproportionate influence on the kinetics of hexamerization, tubulin removal and ultimately severing onset (see Eckert et al. 2012, Fig. 8). As a result, absolute severing times or the fraction of microtubule length remaining at a fixed time point can vary substantially between assays.

Importantly, this assay-to-assay variability was often larger than the difference between microtubules with few versus many defects. For this reason, direct comparison of absolute values across independent experiments (such as between fluorophore-swapping controls) is not possible. To enable a meaningful comparison between microtubules with different defect densities, we therefore immobilized microtubules with few and many defects always side-by-side within the same flow channel, ensuring identical experimental conditions for both microtubule populations. The use of low spastin concentrations was a deliberate experimental choice, allowing us to observe localized spastin-microtubule interactions (Fig. S6) and to prolong the severing process so that differences between individual mechanistic phases could be resolved.

We have now clarified this rationale and the associated assay-to-assay variability when explaining the experimental design in the main text. In addition, we have removed the percentage values of remaining microtubule length at fixed time points and instead focus on comparative trends within the same assay (see also Reviewer 1, point 6).

P8, In figure 2, there is no difference in the severing rate between microtubules with few and many defects at 20 nM spastin. If anything, the microtubules with few defects seem to get cut faster. This

is puzzling and needs explanation.

Reply: If we understand the point correctly, this can be explained by the variability of the assay described in our answer to the previous question. In these assays, we can thus not compare the absolute times, but only the relative times of the two microtubule populations immobilized in the same flow channel. However, despite a slightly faster completion of the total severing process for microtubules with few defects, the average severing onset time was still significantly longer than in those with many defects. **We now comment on the assay-to-assay variability and our strategy to nevertheless perform comparable experiments (see also above).**

P12, Since microtubule severing starts and finishes sooner for ones with more defects, I would have expected a greater decrease in tubulin signal for those microtubules in figure 4b, which is not observed. Specifically, I would have expected a difference between the cyan and magenta curves in the middle panel (rel. Fl of microtubules per μm) after 25 min. Unless I am misunderstanding how this analysis was performed, this discrepancy requires an explanation.

Reply: In fact, after 25 min, we observe a greater decrease in the length-normalized tubulin signal (average fluorescence intensity per μm of microtubule) in microtubules with more defects when we normalize the fluorescence intensity signal to the initial total microtubule length. However, the curve shown in the middle panel of Fig. 4B is normalized to the actual/remaining microtubule length at the respective time point. We decided on this approach because we were not primarily interested in the total tubulin loss but rather in the tubulin loss of the microtubule segments that still appeared intact in the phase prior to severing. We also used the same normalization for the spastin signal. **We now explain more clearly in the manuscript how and why this analysis was performed (in the main text and the legend of Fig. 4B) and additionally provide the average fluorescence intensity of tubulin normalized to the initial microtubule length) in the SI (Fig. S4).**

P16, Line#20 - Provide the make of "airfuge"

Reply: More detailed information about the airfuge was added to the manuscript.

P18, Line#28 - correction needed, perhaps revise to: "Finally, the grids were stained with a 0.75 % uranyl formate solution...."

Reply: The sentence/paragraph was revised for clarity.

Second decision letter

MS ID#: jcs.264497R1

MS Title: Microtubule lattice defects facilitate spastin-mediated severing

Authors: Cordula Reuther; Paula Santos Otte; Rahul Grover; Till Korten; Stefan Diez

Article Type: Research Article

Dear Stefan,

I am happy to tell you that your manuscript has been accepted for publication in Journal of Cell Science, pending standard publication integrity checks.